# Ad vs Organic: Revisiting Incentive Compatible Mechanism Design in E-commerce Platforms

## ABSTRACT

On typical e-commerce platforms, a product can be dislayed to the user in two possible forms, as an ad item or an organic item. Usually ad and organic items are selected separately by the advertising system and recommendation system, and then combined by a merging mechanism. Although the design of the merging mechanism have been extensively studied, little attention has been paid to a critical situation that arises when the set of candidate ad items and organic items overlap. Despite its common occurrence, this situation is not correctly handled by almost all existing works, potentially causing incentive problems for advertisers and violation of economic constraints. To this end, we revisit the design of the merging mechanism. We identify a necessary property called form stability, and provide simplification results of the mechanism design problem. Moreover, we design simple mechanisms strictly ensuring economic properties such as incentive compatibility, and demonstrate that they are approximately optimal under certain assumptions.

## KEYWORDS

E-Commerce, Mechanism Design, Online Advertising, Competitive Ratio

**ACM Reference Format:**
Anonymous Author(s). 2018. Ad vs Organic: Revisiting Incentive Compatible Mechanism Design in E-commerce Platforms. In *Proceedings of Make sure to enter the correct conference title from your rights confirmation emai (Conference acronym 'XX).* ACM, New York, NY, USA, 9 pages. https://doi.org/XXXXXXX.XXXXXXX

## 1 INTRODUCTION

With the rise of e-commerce platforms, business owners now have the opportunity to reach a vast user base and expand their operations through online platforms. E-commerce platforms employ recommendation systems to recommend products to users based on various factors such as user interest. These recommendations play a crucial role in influencing user purchasing decisions and consequently impacting the overall business volume for merchants.

However, business owners often have limited control over the recommendation algorithms, which can hinder their ability to foster business growth. Consequently, they resort to online advertising as an alternative marketing channel. Through strategic bidding on

advertisements, they can effectively promote their products and expand their reach to a broader audience. The online advertising market has experienced remarkable expansion, with global market size reaching 236.90 billion USD in 2022.

On an e-commerce platform, the contents displayed to users consist of organic items and ad items. Organic items are products selected by the recommendation system, while ad items are products selected by the advertising system. Primarily, the recommendation system aims to enhance user experience, while the advertising system aims to maximize advertising revenue. When displayed to users, these two types of items differ in charging and user response. Advertisers are charged by the platform when their ad items are clicked [1], whereas the owners of organic items are not charged. Additionally, the user can usually distinguish between ad and organic items as most e-commerce platforms explicitly label ad items as advertisements, and therefore the response behavior can be different depending on the item type. Users generally show a preference for organic items over ad items. For convenience, when a product is displayed as an ad (or organic) item, we say it is displayed in its ad (or organic) form.

Traditionally, ad and organic items are integrated by designating specific slots in a displayed page for ad items and leaving the rest for organic items. However, this approach is inflexible and potentially inefficient. Recently, there has been significant research on developing a merging mechanism to flexibly merge the ad and organic items. This topic has been explored from various perspectives, including theoretical analysis of optimal design, and practical merging algorithms focusing on modeling the user interests and balancing multiple objectives.

However, we have observed that almost all existing works implicitly rely on a common assumption: the candidate ad items and candidate organic items do not overlap. But this assumption is problematic in real scenarios, as it is quite common for some products to be selected as both ad and organic items. This is because both ad systems and recommendation systems tend to choose the products that closely match user interests or search keywords.

This situation leads to various issues that are not addressed properly by existing works. Firstly, when a product can be displayed in organic form, its owner may obtain a satisfying utility, and will have less incentive to compete in the auction and make the ad form displayed. If the mechanism overlook this phenomenon, the incentive compatibility can be violated. Secondly, when a product is selected as an ad item and an organic item simultaneously, they are usually not allowed to be displayed at the same time, since this would harm user experience and content diversity. Therefore the two forms cannot be treated as two independent items, posing a challenge to the allocation process.

In this paper, we revisit the mechanism design problem under the situation where candidate ad and organic items can overlap, and

---

[1]This is known as the pay-per-click (PPC) mode.

attempt to design and analysis mechanisms that strictly adheres to economic properties such as incentive compatibility. Since the desired economic properties involve the allocation of both ad and organic items, we ignore the original ad auction process in the ad subsystem, and design a content merging mechanism that jointly decide the allocation of all items as well as the prices charged to the advertisers.

Specifically, the content merging mechanism take the candidate ad items and the submitted bids from the advertising system, and take the candidate organic items from the recommendation subsystem. All candidate ad items and candidate organic items are combined together, called the set of candidate products. The content merging mechanism is responsible to make decisions for each candidate product regarding:

(1) Whether this product is displayed to the user, and which slot in the page is allocated to it;
(2) If this product is displayed to the user, the form in which this product is displayed;
(3) If this product is displayed as an ad item, the amount of payment charged to its owner when it is clicked.

In this paper, we examine a single-slot scenario, which is enough to capture the incentive issues. With some assumptions introduced later, we can focus on the case that all candidate products have two display forms of ad and organic, indicating the overlapping situation. Our characterization results demonstrate that the main difficulty in designing the mechanism lies in deciding the display form of each candidate product.

In this paper, we make the following contributions:

- We recognize the incentive problem which arises when the candidate ad and organic items overlap. This highlights a potential flaw that is overlooked in almost all existing works.
- We properly formulate the mechanism design problem under economic constraints. Then we characterize the content merging mechanism under these constraints. We establish that a necessary property called form stability, which help us to further simplify the mechanism design problem.
- We propose two simple incentive compatible mechanisms called FIX and CHANGE, and analyze their performance theoretically. We prove that in the case with two candidate products, CHANGE is optimal, while FIX is $\frac{4}{5}$-competitive relative to the optimal objective. We also prove that when there are $n$ products and all bid distributions are identical, both mechanisms are $\frac{n-1}{n}$-competitive.

## 2 RELATED WORKS

In the field of online advertising and recommender system, there have been extensive research on integrating ad and organic items. A main focus is modeling the mutual influence between different items known as externalities, and developing effective allocation algorithms[2, 5, 7, 13]. This is usually achieved by designing and employing deep learning models and reinforcement learning algorithms. Another main direction is optimizing the displayed outcome considering multiple objectives or constraints, such as the revenue of platform, the utility of advertisers, and the experience of users[3, 4, 8, 9, 11, 12]. A typical approach is to convert the problem

into an unconstrained optimization through Lagrange Duality, and optimize a linear objective. [1, 6] theoretically analyze the optimal allocation under multiple objectives. Theoretical analysis in the field predominantly rely on classical results in auction theory such as the optimal single-parameter auction [10].

To the best of our knowledge, all prior works are implicitly based on the assumption that ad and organic items do not overlap, which can fail in real scenarios. When a product is simultaneously selected as ad and organic items, it is no longer allowed to treat the two candidate items independently, causing possible failure of the existing approaches. Therefore the mechanism design under this situation should be revisited carefully, which is the purpose of our work.

## 3 PRELIMINARY

In this section, we formally define the problem to design the content merging mechanism. To simplify the discussion, we make the assumption that the content merging mechanism can freely determine the display form of a candidate product as ad or organic, regardless of the system that selected it. This causes no negative influence as long as the economic properties are maintained. For products that are not candidate advertisement items, we assume that their owners submit a bid of 0. This, along with the individual rationality constraint, ensures that no payment is charged to the owner even if such a product is displayed as an ad item. In the following discussion, we can assume that all product owners are submitting bids.

Now we give the problem formulation formally. There are $n$ candidate products, denoted by $i \in [n] = \{1, \cdots, n\}$. Each product $i$ can be displayed in two forms, ad or organic. We refer to its two forms as the ad item $\text{Ad}_i$ and the organic item $\text{Org}_i$, respectively. There is a single slot that can display either one ad item or one organic item selected by the content merging mechanism.

The two forms $\text{Ad}_i$ and $\text{Org}_i$ of a product $i$ can be distinguished by the user, and may cause different user feedback. For each product $i \in [n]$, let $\alpha_i^A$ be the click-through rate of its ad form $\text{Ad}_i$ and $\alpha_i^O$ be the click-through rate of its organic form $\text{Org}_i$. We assume $\alpha_i^O > \alpha_i^A$ due to the user's preference for organic items. We also assume that all items' click-through rates are public information.

As previously assumed, the owner of each product $i \in [n]$ (also called owner $i$ for convenience) submits a bid $b_i$. Let $\boldsymbol{b} = (b_1, \cdots, b_n)$ denote the bid profile. Sometimes we use the notation $\boldsymbol{b}_{-i}$ to denote the bid profile submitted by all owners except owner $i$. We assume that each $b_i$ independently follows a prior distribution $F_i$ known to the platform[2]. We further assume that each $F_i$ is a regular distribution with a support of $[0, B_i]$ for some $B_i \geq 0$, which is a standard assumption in economics.

Now we formally define the content merging mechanism. We omit the CTRs and prior distributions in the inputs, since they are known to the platform.

**Definition 3.1** (Content Merging Mechanism). A content merging mechanism is represented by $M = (x, y, p)$, where:

---

[2]By ensuring incentive compatibility, we can assume that all owners are truthfully bidding, and therefore the bid distribution $F_i$ is not influenced by the design of mechanism. We will not delve into this.

(1) $x(\boldsymbol{b}) = (x_i(\boldsymbol{b}))_{i \in [n]}$ and $y(\boldsymbol{b}) = (y_i(\boldsymbol{b}))_{i \in [n]}$ are allocation rules for ad items and organic items, respectively. For each product $i \in [n]$, $x_i(\boldsymbol{b}) = 1$ if its ad form $\text{Ad}_i$ is displayed, and $x_i(\boldsymbol{b}) = 0$ otherwise. Similarly, $y_i(\boldsymbol{b}) = 1$ if its organic form $\text{Org}_i$ is displayed, and $y_i(\boldsymbol{b}) = 0$ otherwise.

(2) $p(\boldsymbol{b}) = (p_i(\boldsymbol{b}))_{i \in [n]}$ denotes the payment charged to the owners when their products are clicked as ads. Specifically, for the owner of product $i$, if $\text{Ad}_i$ is displayed (i.e., $x_i(\boldsymbol{b}) = 1$), then when the user clicks $\text{Ad}_i$ (which happens with probability $\alpha_i^A$), the platform charges a payment of $p_i(\boldsymbol{b})$ to her. We always assume that $p_i(\boldsymbol{b}) = 0$ when $x_i(\boldsymbol{b}) = 0$.

We assume that the product owners are utility maximizers which value the clicks on the ad and organic forms equally. A mechanism $M$ induces two functions $X_i(\boldsymbol{b})$ and $P_i(\boldsymbol{b})$ for each $i \in [n]$, defined as

$$X_i(\boldsymbol{b}) = x_i(\boldsymbol{b})\alpha_i^A + y_i(\boldsymbol{b})\alpha_i^O,$$

and

$$P_i(\boldsymbol{b}) = x_i(\boldsymbol{b})\alpha_i^A p_i(\boldsymbol{b}).$$

Under any (possibly misreporting) bid profile $\boldsymbol{b}$, $X_i(\boldsymbol{b})$ represents the total probability that product $i$ is clicked in either form, and $P_i(\boldsymbol{b})$ represents the expected payment charged to owner $i$. If the truthful bid of owner $i$ is $\tilde{b}_i$, then her utility is defined as $\tilde{b}_i X_i(\boldsymbol{b}) - P_i(\boldsymbol{b})$. The incentive compatibility requires that each owner's utility is always maximized by truthful bidding, which will be formally defined later.

### 3.1 Optimization Objective and Constraints

The platform's objective is to maximize a mixture of revenue and user experience. The revenue target is defined as the expected total payment

$$\text{Rev}(M) := \mathbb{E}_{\boldsymbol{b} \sim F}\left[\sum_{i \in [n]} x_i(\boldsymbol{b})\alpha_i^A p_i(\boldsymbol{b})\right]. \tag{1}$$

For user experience, we assume that each ad item $\text{Ad}_i$ has a user experience effect $\gamma_i^A$, and each organic item $\text{Org}_i$ has a user experience effect $\gamma_i^O$. We assume that $0 \le \gamma_i^A \le \gamma_i^O$ due to user's preference for organic items. The user experience target is defined as the expected user experience

$$\text{UE}(M) := \mathbb{E}_{\boldsymbol{b} \sim F}\left[\sum_{i \in [n]} \left(x_i(\boldsymbol{b})\gamma_i^A + y_i(\boldsymbol{b})\gamma_i^O\right)\right]. \tag{2}$$

Without loss of generality, we assume that

$$\text{OBJ}(M) := \text{Rev}(M) + \text{UE}(M) \tag{3}$$

Note that this formulation fully captures any weighted combination of the revenue and user experience targets, since the weights can be encoded into $\gamma_i^A$ and $\gamma_i^O$.

The content merging mechanism should satisfy the following constraints:

- Feasibility: Only one product is displayed. Formally, for any $\boldsymbol{b}$,

$$\sum_{i \in [n]} (x_i(\boldsymbol{b}) + y_i(\boldsymbol{b})) \le 1.$$

We make the free disposal assumption that if $\sum_{i \in [n]}(x_i(\boldsymbol{b}) + y_i(\boldsymbol{b})) = 0$, i.e., no item is selected to display, then some neutral content is displayed to the user which has no effect on user experience.

- Incentive Compatibility (IC): Each owner has no incentive to misreport the bid. Formally, for any $\boldsymbol{b}$ and $i \in [n]$, for any misreporting bid $b_i'$,

$$b_i X_i(b_i, \boldsymbol{b}_{-i}) - P_i(b_i, \boldsymbol{b}_{-i}) \ge b_i X_i(b_i', \boldsymbol{b}_{-i}) - P_i(b_i', \boldsymbol{b}_{-i}).$$

- Individual Rationality (IR): The payments never exceed the bids, which guarantees that an owner's utility is always non-negative. Formally, for any $\boldsymbol{b}$ and any $i \in [n]$,

$$p_i(\boldsymbol{b}) \le b_i.$$

The content merging problem is to maximize Equation (3) under the above constraints. Formally,

$$\begin{aligned} \max_{\mathcal{M}} \quad & \text{OBJ}(\mathcal{M}) \\ s.t. \quad & \text{IC} \\ & \text{IR} \\ & \text{Feasibility} \end{aligned} \tag{4}$$

We denote the optimal solution to the above optimization problem by $\mathcal{M}^{OPT}$. We call a mechanism $\mathcal{M}$ is $\tau$-competitive if the ratio between $OBJ(\mathcal{M})$ and $OBJ(\mathcal{M}^{OPT})$ is at least $\tau$.

## 4 CHARACTERIZATION ON INCENTIVE COMPATIBLE MECHANISM

In this section we charcterize the content merging mechanism under the constraints of IC, IR, and Feasibility. Based on the seminal work of Myerson [10] in auction theory, we show that the payments are uniquely determined and the IC constraint is turned into a monotonicity constraint on the allocation rules. Additionally, we observe a property implied by the constraints, called form stability, which means that a product's form is unchanged when only its owner modifies the bid. This property helps us to further simplify the design of the content merging mechanism.

### 4.1 Characterizing Incentive Compatibility and Revenue

By adapting Myerson's theory [10], we have the following two lemmata:

**Lemma 4.1** ([10]). *A content merging mechanism $\mathcal{M}$ satisfies IC and IR if and only if the following statements hold:*

(1) *(Allocation Monotonicity) For any product $i \in [n]$, given the bids $\boldsymbol{b}_{-i}$ of other products, its click probability is non-decreasing in its bid $b_i$. Formally, for any $b_i$ and $b_i'$ that $b_i < b_i'$, it holds that*

$$X_i(b_i, \boldsymbol{b}_{-i}) \le X_i(b_i', \boldsymbol{b}_{-i}).$$

(2) *(Payment Identity) For any product $i \in [n]$, it holds for any bid profile $\boldsymbol{b}$ that*

$$P_i(\boldsymbol{b}) = b_i \cdot X_i(b_i, \boldsymbol{b}_{-i}) - \int_0^{b_i} X_i(t, \boldsymbol{b}_{-i})dt,$$

*where $X_i(\boldsymbol{b})$ and $P_i(\boldsymbol{b})$ are previously defined at (??) and (??), respectively.*

**Lemma 4.2** ([10]). *If $\mathcal{M}$ satisfies IC and IR, then the revenue target can be written as*

$$\text{Rev}(\mathcal{M}) = \mathop{\mathbb{E}}_{\boldsymbol{b} \sim F} [X_i(\boldsymbol{b})\phi_i(b_i)],$$

*where $\phi_i(b_i)$ is defined as $\phi_i(b_i) = b_i - (1 - F_i(b_i))/f_i(b_i)$, known as Myerson's virtual value function. Here $F_i(b_i)$ and $f_i(b_i)$ denotes the cumulative density function and probability density function of the distribution $F_i$, respectively.*

With Lemma 4.1, recall that $X_i(\boldsymbol{b}) = x_i(\boldsymbol{b})\alpha_i^A + y_i(\boldsymbol{b})\alpha_i^O$ and $P_i(\boldsymbol{b}) = x_i(\boldsymbol{b})\alpha_i^A p_i(\boldsymbol{b})$. One can see that $y_i(\boldsymbol{b})$ appears only in the definition of $X_i(\boldsymbol{b})$, and not in $P_i(\boldsymbol{b})$. Consequently, $y_i(\boldsymbol{b})$ influences the right side of Payment Identity, but cannot affect the left side. This indicates that, although the allocation of an organic item provides utility to its owner, the mechanism cannot charge for it in order to balance the owner's incentive. This phenomenon will be signified if we extend our setting to more general scenarios such as the multi-slot case. In comparison, in the setting of existing works where ad items and organic items do not overlap, $y_i(\boldsymbol{b})$ can always be set as 0 for each advertiser, and therefore does not cause incentive problems. This distinction sets our scenario apart from traditional single-parameter settings, making the conventional approaches to design optimal mechanisms inapplicable.

Due to the special influence of organic items on the incentive of product owners, we have the following corollary, which will imply the property of form stability introduced later.

**Corollary 4.3.** *Suppose $\mathcal{M}$ satisfies IC, IR, and Feasibility. For any bid profile $\boldsymbol{b}$ and any $i \in [n]$, if there is $j \in [m]$ such that $y_{i,j}(\boldsymbol{b}) = 1$, i.e., $\text{Org}_i$ is displayed, then for any $b_i' \in (0, b_i)$, it must hold that*

$$X_i(b_i', \boldsymbol{b}_{-i}) = X_i(b_i, \boldsymbol{b}_{-i}).$$

PROOF. By Feasibility constraint, we know that $\text{Ad}_i$ is not displayed under $\boldsymbol{b}$, i.e., for any $j \in [m]$, $x_{i,j}(\boldsymbol{b}) = 0$. By (??), it follows that $P_i(\boldsymbol{b}) = 0$. By the Payment Identity in Lemma 4.1, we have $b_i \cdot X_i(b_i, \boldsymbol{b}_{-i}) - \int_0^{b_i} X_i(t, \boldsymbol{b}_{-i})dt = P_i(\boldsymbol{b}) = 0$, i.e., $\int_0^{b_i}(X_i(b_i, \boldsymbol{b}_{-i}) - X_i(t, \boldsymbol{b}_{-i}))dt = P_i(\boldsymbol{b}) = 0$. And by the Allocation Monotonicity in Lemma 4.1, $X_i(t, \boldsymbol{b}_{-i})$ is non-decreasing in $t$. For any $b_i' \in (0, b_i)$, suppose for contradiction that $X_i(b_i', \boldsymbol{b}_{-i}) < X_i(b_i, \boldsymbol{b}_{-i})$, then by the monotonicity we have $\int_0^{b_i}(X_i(b_i, \boldsymbol{b}_{-i}) - X_i(t, \boldsymbol{b}_{-i}))dt \geq b_i'(X_i(b_i, \boldsymbol{b}_{-i}) - X_i(b_i', \boldsymbol{b}_{-i})) > 0$, which contradicts. Therefore $X_i(b_i', \boldsymbol{b}_{-i}) = X_i(b_i, \boldsymbol{b}_{-i})$. □

## 4.2 Form Stability

We observe a property called form stability, which holds for any mechanism satisfying IC, IR and feasibility. It roughly means that a product's form remains unchanged when only its owner modifies her bid. And equivalently, whether a product's organic form $\text{Org}_i$ is displayed is not influenced by $b_i$.

**Lemma 4.4.** *If the content merging mechanism $\mathcal{M}$ satisfies IC, IR and Feasibility, then the following form stability holds: For any $i \in [n]$, for any $\boldsymbol{b}_{-i}, b_i, b_i'$,*

$$y_i(b_i, \boldsymbol{b}_{-i}) = 1 \implies x_i(b_i', \boldsymbol{b}_{-i}) = 0.$$

PROOF. We prove by contradiction. Suppose that form stability does not hold for some product $i \in [n]$. It means that under some $\boldsymbol{b}_{-i}$, there exist $b_i$ and $b_i'$, such that $y_i(b_i, \boldsymbol{b}_{-i}) = 1$ and $x_i(b_i', \boldsymbol{b}_{-i}) = 1$. Therefore $X_i(b_i, \boldsymbol{b}_{-i}) = \alpha_i^O$ and $X_i(b_i', \boldsymbol{b}_{-i}) = \alpha_i^A$, and we have $X_i(b_i, \boldsymbol{b}_{-i}) > X_i(b_i', \boldsymbol{b}_{-i})$ due to the assumption $\alpha_i^O > \alpha_i^A$. By Allocation Monotonicity in Lemma 4.1, it must hold that $b_i > b_i'$. However, since $\text{Org}_i$ is displayed under the bid profile $(b_i, \boldsymbol{b}_{-i})$, by Corollary 4.3 we have $X_i(b_i, \boldsymbol{b}_{-i}) = X_i(b_i', \boldsymbol{b}_{-i})$, which contradicts. □

And this is basically equivalent to the following:

**Lemma 4.5.** *Suppose $M$ satisfies IC, IR, Feasibility, and consequently form stability. Then for any $i \in [n]$, $j \in [m]$, $y_i(\boldsymbol{b})$ only depends on $\boldsymbol{b}_{-i}$. Formally, for any $\boldsymbol{b}_{-i}$, for any $b_i \neq b_i'$, we have*

$$y_i(b_i, \boldsymbol{b}_{-i}) = y_i(b_i', \boldsymbol{b}_{-i}).$$

PROOF. Suppose for contradiction that there exist $i \in [n]$, $\boldsymbol{b}_{-i}$, $b_i$ and $b_i'$, such that $y_i(b_i, \boldsymbol{b}_{-i}) \neq y_i(b_i', \boldsymbol{b}_{-i})$. Without loss of generality, assume that $y_i(b_i, \boldsymbol{b}_{-i}) = 1$ and $y_i(b_i', \boldsymbol{b}_{-i}) = 0$. We can prove that $X_i(b_i', \boldsymbol{b}_{-i}) > 0$: If $b_i' < b_i$, by Corollary 4.3 we have $X_i(b_i', \boldsymbol{b}_{-i}) = X_i(b_i, \boldsymbol{b}_{-i}) > 0$. And if $b_i' > b_i$, by Lemma 4.1 we have $X_i(b_i', \boldsymbol{b}_{-i}) \geq X_i(b_i, \boldsymbol{b}_{-i}) > 0$. However, by form stability, we have $x_i(b_i', \boldsymbol{b}_{-i}) = 0$, and therefore $X_i(b_i', \boldsymbol{b}_{-i}) = 0$, which contradicts. □

With form stability, we further simplify the problem. Firstly we can observe that a product's organic form is no longer involved in the payment computation of its ad form.

**Lemma 4.6.** *Suppose that $M$ satisfies IC, IR and Feasibility, and consequently form stability. Under any $\boldsymbol{b}$, if $\text{Ad}_i$ is displayed, then $p_i(\boldsymbol{b})$ is given by*

$$p_i(\boldsymbol{b}) = \inf\{b_i' \geq 0 : x_i(b_i', \boldsymbol{b}_{-i}) = 1\},$$

*which is the critical bid for $\text{Ad}_i$ to be displayed.*

PROOF. If $\text{Ad}_i$ is displayed, i.e., $x_i(\boldsymbol{b}) = 1$, by feasibility we have $y_i(\boldsymbol{b}) = 0$, which implies that $y_i(\boldsymbol{b}) = 0$ By Lemma 4.5, we have $y_i(b_i', \boldsymbol{b}_{-i})$ for all $b_i'$. Therefore $X_i(b_i', \boldsymbol{b}_{-i}) = x_i(b_i', \boldsymbol{b}_{-i})\alpha_i^A$. By Allocation Motononicity in Lemma 4.1, we know that $x_i(b_i', \boldsymbol{b}_{-i})$ is non-decreasing in $b_i'$. Let $\hat{b} = \inf\{b_i' \geq 0 : x_i(b_i', \boldsymbol{b}_{-i}) = 1\}$, then $x_i(b_i', \boldsymbol{b}_{-i}) = 0$ for all $b_i' \in (0, \hat{b})$, and $x_i(b_i', \boldsymbol{b}_{-i}) = 1$ for all $b_i' > \hat{b}$. Therefore by Payment Identity in Lemma 4.1 we can calculate that $P_i(b_i', \boldsymbol{b}_{-i}) = \hat{b}\alpha_i^A$ for all $b_i' > \hat{b}$, and it follows that $p_i(b_i', \boldsymbol{b}_{-i}) = \hat{b}$, which is the desired result. □

Moreover, the revenue also no longer involve the allocation of organic items.

**Lemma 4.7.** *If $\mathcal{M}$ satisfies IC, IR, Feasibility and form stability, then the revenue target can be written as*

$$\text{Rev}(\mathcal{M}) = \mathop{\mathbb{E}}_{\boldsymbol{b} \sim F} \left[ \sum_{i \in [n]} x_i(\boldsymbol{b})\alpha_i^A \phi_i(b_i) \right],$$

*where $\phi_i(b_i)$ is the Myerson's virtual value function.*

PROOF. By Lemma 4.2, we have

$$\text{Rev}(\mathcal{M}) = \mathop{\mathbb{E}}_{\boldsymbol{b} \sim F}\left[\sum_{i \in [n]} X_i(\boldsymbol{b})\phi_i(b_i)\right]$$

$$= \sum_{i \in [n]} \mathop{\mathbb{E}}_{\boldsymbol{b} \sim F}[X_i(\boldsymbol{b})\phi_i(b_i)\mathbb{I}[y_i(\boldsymbol{b}) = 1]] +$$

$$\sum_{i \in [n]} \mathop{\mathbb{E}}_{\boldsymbol{b} \sim F}[X_i(\boldsymbol{b})\phi_i(b_i)\mathbb{I}[y_i(\boldsymbol{b}) = 0]])$$

For any $i \in [n]$, observe that $\mathbb{E}_{b_i \sim F_i}[\phi_i(b_i)] = \int_0^B (t - \frac{1-F_i(t)}{f_i(t)})f_i(t)dt = \int_0^B tf_i(t)dt - \int_0^B \frac{1-F_i(t)}{d}t = E[b_i] - E[b_i] = 0$. For any $\boldsymbol{b}_{-i}$, by Lemma 4.5, if there exists $b_i$ such that $y_i(b_i, \boldsymbol{b}_{-i}) = 1$, then it holds for all $b_i'$ that $y_{i,j}(b_i', \boldsymbol{b}_{-i}) = 1$. That is, $\mathbb{I}[y_i(\boldsymbol{b}) = 1] = \mathbb{I}[\exists b_i', y_i(b_i', \boldsymbol{b}_{-i}) = 1]$. It follows that

$$\mathop{\mathbb{E}}_{\boldsymbol{b} \sim F}[X_i(\boldsymbol{b})\phi_i(b_i)\mathbb{I}[y_i(\boldsymbol{b}) = 1]]$$

$$= \mathop{\mathbb{E}}_{\boldsymbol{b}_{-i} \sim F_{-i}}\left[\alpha_i^O \mathbb{I}[\exists b_i', y_i(b_i', \boldsymbol{b}_{-i}) = 1] \cdot \mathop{\mathbb{E}}_{b_i \sim F_i}[\phi_i(b_i)]\right]$$

$$= 0.$$

Therefore, we have

$$\text{Rev}(\mathcal{M}) = \mathop{\mathbb{E}}_{\boldsymbol{b} \sim F}\left[\sum_{i \in [n]} X_i(\boldsymbol{b})\phi_i(b_i)\right]$$

$$= \sum_{i \in [n]} \mathop{\mathbb{E}}_{\boldsymbol{b} \sim F}[X_i(\boldsymbol{b})\phi_i(b_i)\mathbb{I}[y_i(\boldsymbol{b}) = 0]])$$

$$= \sum_{i \in [n]} \mathop{\mathbb{E}}_{\boldsymbol{b} \sim F}\left[x_i(\boldsymbol{b})\alpha_i^A \phi_i(b_i)\right]$$

$$= \mathop{\mathbb{E}}_{\boldsymbol{b} \sim F}\left[\sum_{i \in [n]} x_i(\boldsymbol{b})\alpha_i^A \phi_i(b_i)\right].$$

□

By Lemma 4.6, we know that $p_i(\boldsymbol{b})$ is uniquely determined by $x_i(\boldsymbol{b})$. And by Lemma 4.7, we can write the revenue target into a linear form of $x_i(\boldsymbol{b})$. Therefore, when $x(\boldsymbol{b})$ and $y(\boldsymbol{b})$ are given, a unique content merging mechanism $\mathcal{M}^{x,y}$ is specified, and the objective function $\text{OBJ}(\mathcal{M}^{x,y})$ can be expressed using $x(\boldsymbol{b})$ and $y(\boldsymbol{b})$. Formally, the content merging problem (4) can be rewritten as the following optimization problem:

$$\max_{x,y} \quad \text{OBJ}(\mathcal{M}^{x,y}) = \mathop{\mathbb{E}}_{\boldsymbol{b} \sim \mathcal{F}}[\sum_{i \in [n]}(a_i x_i(\boldsymbol{b}) + o_i y_i(\boldsymbol{b}))]$$

$$s.t. \quad \sum_{i \in [n]}(x_i(\boldsymbol{b}) + y_i(\boldsymbol{b})) \le 1, \forall \boldsymbol{b},$$

$$y_i(b_i, \boldsymbol{b}_{-i}) = y_i(b_i', \boldsymbol{b}_{-i}), \forall i \in [n], \boldsymbol{b}_{-i}, b_i, b_i',$$

$$x_i(b_i, \boldsymbol{b}_{-i}) \le x_i(b_i', \boldsymbol{b}_{-i}), \forall i \in [n], \boldsymbol{b}_{-i}, b_i < b_i'.$$

where $a_i = \phi_i(b_i)\alpha_i^A + \gamma_i^A$ and $o_i = \gamma_i^O$ denote the contribution of $\text{Ad}_i$ and $\text{Org}_i$ to the objective function, respectively. This allows us to omit $p_i(\boldsymbol{b})$ and focus on designing the allocation rules.

By the free-disposal assumption, without loss of generality we can assume that all $a_i$ and $o_i$ are truncated to be non-negative. For each product $i$, the distribution of $b_i$ (i.e., $F_i$) induces a distribution

of $a_i$, which we denote as $G_i$. We also define $G = \times_{i \in [n]}G_i$ as the product distribution of $G_i$. Since $F_i$ is assumed to be regular, we know that $\phi_i(b_i)$ is monotone increasing, and $G_i$ has no mass point except 0. From now on, we can use $\boldsymbol{a} = (a_1, \cdots, a_n)$ as the proxy of $\boldsymbol{b}$. With slight abuse of notations, we use $x(\boldsymbol{a}) = (x_i(\boldsymbol{a}))_{i \in [n]}$ and $y(\boldsymbol{a}) = (y_i(\boldsymbol{a}))_{i \in [n]}$ to denote the allocation of ad and organic items when given $\boldsymbol{a}$ instead of $\boldsymbol{b}$. The problem is then rewritten as:

$$\max_{x,y} \text{OBJ}(\mathcal{M}^{x,y}) = \mathop{\mathbb{E}}_{\boldsymbol{a} \sim G}[\sum_{i \in [n]}(a_i x_i(\boldsymbol{a}) + o_i y_i(\boldsymbol{a}))]$$

$$s.t. \quad \sum_{i \in [n]}(x_i(\boldsymbol{a}) + y_i(\boldsymbol{a})) \le 1, \forall \boldsymbol{a}, \quad (5)$$

$$y_i(a_i, \boldsymbol{a}_{-i}) = y_i(a_i', \boldsymbol{a}_{-i}), \forall i \in [n], \boldsymbol{a}_{-i}, a_i, a_i',$$

$$x_i(a_i, \boldsymbol{a}_{-i}) \le x_i(a_i', \boldsymbol{a}_{-i}), \forall i \in [n], \boldsymbol{a}_{-i}, a_i < a_i'.$$

With a little abuse of notations, we also call this optimization problem to be content merging problem, and such $\mathcal{M}$ to be content merging mechanism.

## 5 TWO SIMPLE CONTENT MERGING MECHANISMS

In this section, we give two content merging mechanisms that strictly adhere to the economic properties, and analyze their performance under certain conditions. Following the problem simplification given in the last section, we only need to design $x(\boldsymbol{a})$ and $y(\boldsymbol{a})$ for the optimization problem (5).

### 5.1 FIX Mechanism

Firstly we define the FIX mechanism, which is the best mechanism satisfying that the display form of each product is fixed, i.e., the mechanism only considers one of $\text{Org}_i$ and $\text{Ad}_i$, and discard the other.

We start with the mechanism where all products are always treated as ad items. In this case, we only need to sort all the products according to $a_i$'s and choose the largest one.

**Definition 5.1** (Advertising-Only Mechanism). The advertising-only mechanism, denoted by $\mathcal{M}_0$, is given by

$$x_i(\boldsymbol{a}) = \begin{cases} 1 & \text{if } a_i > \max\{a_j : jj \in [n] \setminus \{i\}\}, \\ 0 & \text{otherwise.} \end{cases}$$

$$y_i(\boldsymbol{a}) = 0.$$

We have

$$\text{OBJ}(\mathcal{M}_0) = \mathop{\mathbb{E}}_{\boldsymbol{a} \sim G}\left[\max_i a_i\right]. \quad (6)$$

Observe that if we ignore the organic items, i.e., assuming $o_i \equiv 0$, then mechanism $\mathcal{M}_0$ is the optimal solution to the optimization problem (5). Actually, it coincides with the common approach to sort the items by some ranking score, in existing works handling multiple optimization objectives such as [1].

However, organic items can possibly have better user experience effects than ad items, it is not reasonable to ignore them. Therefore, we are able to improve the advertising-only mechanism by switching some products from the ad form to the organic form. Since there is only one slot, and all $o_i$'s are known to the mechanism, it is enough to switch one product to the organic form. Specifically,

the mechanism fix the display form of some product $k$ to organic, sort all the products according to $\{a_1, \cdots a_{k-1}, o_k, a_{k+1}, \cdots\}$, and choose the largest.

**Definition 5.2** (FIX-$k$ Mechanism). The FIX-$k$ mechanism, denoted by $\mathcal{M}_k^F$ where $k \in [n]$, is given by

$$x_i(\boldsymbol{a}) = \begin{cases} 1 & i \neq k \text{ and } a_i > \max\{a_j : j \in [n] \setminus \{i, k\}\} \text{ and } a_i > o_k, \\ 0 & \text{otherwise.} \end{cases}$$

$$y_i(\boldsymbol{a}) = \begin{cases} 1 & i = k \text{ and } o_k > \max\{a_j : j \in [n] \setminus \{k\}\}, \\ 0 & \text{otherwise.} \end{cases}$$

For the FIX-$k$ mechanism, we have

$$\text{OBJ}(\mathcal{M}_k^F) = \mathop{\mathbb{E}}_{a_{-k} \sim G_{-k}} \left[ \max\{o_k, \max_{i \neq k} a_i\} \right]. \tag{7}$$

It is easy to verify that the constraints in (5) holds for the allocation rules of these mechanisms, so all of them satisfy IC, IR and feasibility. The FIX mechanism is defined as the best mechanism among these mechanisms.

**Definition 5.3** (FIX Mechanism). The FIX mechanism, denoted by $\mathcal{M}^F$, runs the mechanism with the highest objective value in $\mathcal{M}_0, \mathcal{M}_1^F, \mathcal{M}_2^F, \cdots, \mathcal{M}_n^F$.

**Theorem 5.4.** *In the case with $n = 2$, $\mathcal{M}^F$ is $\frac{4}{5}$-competitive relative to the optimal mechanism.*

PROOF OF THEOREM 5.4. When $n = 2$, $\mathcal{M}^F$ runs the best mechanism among $\mathcal{M}_0, \mathcal{M}_1^F, \mathcal{M}_2^F$. We have that

$$\text{OBJ}(\mathcal{M}_0) = \mathop{\mathbb{E}}_{a_1, a_2} [\max\{a_1, a_2\}],$$

$$\text{OBJ}(\mathcal{M}_1^F) = \mathop{\mathbb{E}}_{a_2} [\max\{o_1, a_2\}],$$

$$\text{OBJ}(\mathcal{M}_2^F) = \mathop{\mathbb{E}}_{a_1} [\max\{a_1, o_2\}].$$

Let $x^*(\boldsymbol{a}), y^*(\boldsymbol{a})$ denote the optimal solution of the optimization problem (5), which induces the optimal mechanism $\mathcal{M}^*$. Let $\text{OBJ}^*(\boldsymbol{a}) = \sum_{i \in [n]} (a_i x_i^*(\boldsymbol{a}) + o_i y_i^*(\boldsymbol{a}))$ denote the objective obtained by $\mathcal{M}^*$ given certain $\boldsymbol{a}$.

Firstly we prove that either $y_1^*(\boldsymbol{a}) \equiv 0$, or $y_2^*(\boldsymbol{a}) \equiv 0$. Suppose for contradiction that there exists $a_1, a_2, a_1', a_2'$ such that $y_1^*(a_1, a_2) = 1$ and $y_2^*(a_1', a_2') = 1$, then by form stability we have $y_1^*(a_1', a_2) = 1$ and $y_2^*(a_1', a_2) = 1$, which contradicts with the feasibility constraint.

By symmetry, we can assume that $y_1^*(\boldsymbol{a}) \equiv 0$. By form stability there is a function $\hat{y}_2^*(a_1)$ such that for any $\boldsymbol{a} = (a_1, a_2), y_2^*(a_1, a_2) = \hat{y}_2^*(a_1)$. For any $a_1$, if $y_2(a_1) = 0$, we have $\text{OBJ}^*(\boldsymbol{a}) \leq \max\{a_1, a_2\}$. And if $y_2(a_1) = 1$, we have $\text{OBJ}^*(\boldsymbol{a}) = o_2 \leq \max\{a_1, o_2\}$. Intuitively, $\mathcal{M}^*$ can decide to follow either $\mathcal{M}_0$ or $\mathcal{M}_2^F$ based on the value of $a_1$. Thus we have

$$\text{OBJ}(\mathcal{M}^*) \leq \mathop{\mathbb{E}}_{a_1 \sim G_1} \left[ \max \left\{ \mathop{\mathbb{E}}_{a_2 \sim G_2} [\max\{a_1, a_2\}], \max\{a_1, o_2\} \right\} \right], \tag{8}$$

We assume w.l.o.g. that $o_1 = 0$ and $o_2 = 1$, and define $h_1(t) = \max\{t, o_2\} = \max\{t, 1\}$, $h_2(t) = \mathbb{E}_{a_2 \sim G_2} [\max\{t, a_2\}]$. We have

$$\frac{\text{OBJ}(\mathcal{M}^F)}{\text{OBJ}(\mathcal{M}^{OPT})}$$

$$\geq \frac{\max\{\text{OBJ}(\mathcal{M}_0), \text{OBJ}(\mathcal{M}_2^F)\}}{\text{OBJ}(\mathcal{M}^{OPT})}$$

$$\geq \frac{\max\{\mathbb{E}_{a_1 \sim G_1} [h_1(a_1)], \mathbb{E}_{a_1 \sim G_1} [h_2(a_1)]\}}{\mathbb{E}_{a_1 \sim G_1} [\max\{h_1(a_1), h_2(a_1)\}]}.$$

To derive an lower bound on this, define $\rho(t) = \frac{h_2(t)}{h_1(t)}$, we investigate the range of $\rho(t)$.

We can calculate the derivatives

$$h_1'(t) = \begin{cases} 0 & t < 1, \\ 1 & t > 1. \end{cases}$$

$$h_2'(t) = \mathbb{P}(a_2 \leq t) \geq 0.$$

When $t \in [0, 1)$, we have

$$\rho'(t) = \frac{h_2'(t) h_1(t) - h_1'(t) h_2(t)}{h_1^2(t)} = h_2'(t) \geq 0.$$

When $t > 1$,

$$\rho'(t) = \frac{t \cdot \mathbb{P}(a_2 \leq t) - \mathbb{E}_{a_2 \sim G_2} [\max\{t, a_2\}]}{h_1^2(t)} \leq 0.$$

Thus, $\rho(t)$ takes the maximum value at $t = 1$, with $\rho(1) = h_2(1) = \mathbb{E}_{a_2} [\max\{1, a_2\}]$.

If $h_2(0) > h_1(0) = 1$, then $h_2(t) \geq t$ and $h_2(t) \geq 1$ holds for all $t \geq 0$, and therefore $h_2(t) \geq h_1(t)$ always holds, implying that $\frac{\text{OBJ}(\mathcal{M}^F)}{\text{OBJ}(\mathcal{M}^{OPT})} = 1$. Therefore we only need to consider the case that $h_2(0) \leq h_1(0)$. In this case, the minimum value of $\rho(t)$ is $\rho(0) = \mathbb{E}[a_2]$.

Observe that $\rho(1) = \mathbb{E}_{a_2} [\max\{1, a_2\}] \leq \rho(0) + 1$. Define $r := \rho(0)$, then it holds for any $t \in [0, +\infty)$ that $\frac{h_2(t)}{h_1(t)} = \rho(t) \in [r, r + 1]$.

We use the following claim, which is proved in the appendix:

**Lemma 5.5.** *If $\frac{h_2(t)}{h_1(t)} \in [c, d]$ for all $t$, where $0 < c < 1 < d$, then for any distribution $G_1$, $\frac{\max\{\mathbb{E}_{a_1 \sim G_1} [h_1(a_1)], \mathbb{E}_{a_1 \sim G_1} [h_2(a_1)]\}}{\mathbb{E}_{a_1 \sim G_1} [\max\{h_1(a_1), h_2(a_1)\}]} \geq \frac{d-c}{2d-1-cd}$.*

Since $\frac{h_2(t)}{h_1(t)} \in [r, r + 1]$, by this lemma, we have that

$$\frac{\max\{\mathbb{E}_{a_1 \sim G_1} [h_1(a_1)], \mathbb{E}_{a_1 \sim G_1} [h_2(a_1)]\}}{\mathbb{E}_{a_1 \sim G_1} [\max\{h_1(a_1), h_2(a_1)\}]}$$

$$\geq \frac{r + 1 - r}{2(r + 1) - 1 - r(r + 1)}$$

$$= \frac{1}{-r^2 + r + 1}$$

$$= \frac{1}{-(r - \frac{1}{2})^2 + \frac{5}{4}}$$

$$\geq \frac{4}{5}.$$

That is, $\mathcal{M}^F$ is $\frac{4}{5}$-competitive relative to the optimal mechanism when $n = 2$. To show the tightness of this ratio, we can construct

the worst case as follows:

$$P(a_1 = 0) = 1/2, \qquad P(a_1 = 1) = 1/2,$$

$$P(a_2 = 0) = 1 - \epsilon, \qquad P(a_2 = \frac{1}{2\epsilon}) = \epsilon,$$

where $\epsilon > 0$ is sufficiently small.

Recall that $o_1 = 0, o_2 = 1$. The optimal mechanism $\mathcal{M}^*$ is given by $y_2^*(0,0) = y_2^*(0, \frac{1}{2\epsilon}) = 1, x_1^*(1,0) = 1, x_2^*(1, \frac{1}{2\epsilon}) = 1$, and all other values are 0. Then we can calculate that $\mathrm{OBJ}(\mathcal{M}^*) = \frac{1}{2} \cdot 1 + \frac{1}{2}((1 - \epsilon) \cdot 1 + \epsilon \frac{1}{2\epsilon}) = \frac{5}{4} - \frac{\epsilon}{2}, \mathrm{OBJ}(\mathcal{M}^F) = \max\{\mathrm{OBJ}(\mathcal{M}_0), \mathrm{OBJ}(\mathcal{M}_2^F)\} = \max\{1, 1 - \frac{\epsilon}{2}\} = 1$. When $\epsilon \to 0$, we have

$$\frac{\mathrm{OBJ}(\mathcal{M}^F)}{\mathrm{OBJ}(\mathcal{M}^*)} \to \frac{4}{5}.$$

□

Besides the $n = 2$ case, we deduce a competitive ratio result for general case, which states as follows,

**Theorem 5.6.** *For content merging problem with $n$ participants, assume the bid distributions are identical, $\mathcal{M}^F$ is $\frac{n-1}{n}$ competitive relative to the optimal mechanism $\mathcal{M}^*$.*

**Proof.** Firstly, observe that the objective of the optimal mechanism $\mathcal{M}^*$ is always bounded by $\max\{o_1, \cdots, o_n, a_1, \cdots, a_n\}$. It follows that

$$\mathrm{OBJ}(\mathcal{M}^*) \leq \underset{a \sim G}{\mathbb{E}} [\max\{o_1, \cdots, o_n, a_1, \cdots, a_n\}]$$

$$= \int_0^\infty \mathbb{P}_{a \sim G}(\max\{o_1, \cdots, o_n, a_1, \cdots, a_n\} \geq t)dt$$

Without loss of generality, assume that $o_1 = \max\{o_1, \cdots, o_n\}$. Since all bid distributions are identical, all $a_i$'s follows the identical distribution denoted by $G_0$. For any $t \geq 0$, it holds that

$$\mathbb{P}_{a \sim G}(\max\{o_1, \cdots, o_n, a_1, \cdots, a_n\} \geq t) = \begin{cases} 1, & t \leq o_1, \\ 1 - G_0(t)^n, & t > 1. \end{cases}$$

where $G_0(t)$ is the cumulative density function of $G_0$. Therefore we have

$$\mathrm{OBJ}(\mathcal{M}^*) \leq o_1 + \int_{o_1}^\infty (1 - G_0(t)^n)dt.$$

Next, we consider the FIX mechanism $\mathcal{M}^F$. We have that

$$\mathrm{OBJ}(\mathcal{M}^F) \geq \mathrm{OBJ}(\mathcal{M}_1^F)$$

$$= \underset{a_{-1} \sim G_{-1}}{\mathbb{E}} \left[ \max\{o_1, \max_{i \neq 1} a_i\} \right]$$

$$= \int_0^\infty \mathbb{P}_{a_{-1} \sim G_{-1}}(\max\{o_1, \max_{i \neq 1} a_i\} \geq t)dt$$

$$= o_1 + \int_{o_1}^\infty (1 - G_0(t)^{n-1})dt.$$

Observe that fixed any $t$, $1 - G_0(t)^n$ can be viewed as a concave function in $n$, thus we have $1 - G_0(t)^{n-1} \geq \frac{n-1}{n}(1 - G_0(t)^n) + \frac{1}{n}(1 - G_0(t)^0) = \frac{n-1}{n}(1 - G_0(t)^n)$. Thus, we have $\mathrm{OBJ}(\mathcal{M}^F) \geq \frac{n-1}{n}\mathrm{OBJ}(\mathcal{M}^*)$, that is,

$$\frac{OBJ(\mathcal{M}^F)}{OBJ(\mathcal{M}^{OPT})} \geq \frac{n-1}{n}.$$

□

## 5.2 The CHANGE Mechanism

Although the FIX mechanism $\mathcal{M}^F$ takes the organic items into consideration, but the mechanism is of limited flexibility to some extent. Inspired by the optimal mechanism in the case of $n = 2$, although the display form of product $k$ should not be influenced by $a_k$ due to form stability, it can be decided according to the value of $a_{-k}$. Intuitively, given the value of the other ad forms, we compare the expected objective obtained by using $\mathrm{Org}_k$ and $\mathrm{Ad}_k$, and choose the better one. This gives the following CHANGE-$k$ mechanism.

**Definition 5.7** (CHANGE-$k$ mechanism). The CHANGE-$k$ mechanism, denoted by $\mathcal{M}_k^C$ where $k \in [n]$, is given as follows:

For any $a$, define $w_k^A(a_{-k}) = \mathbb{E}_{a_k' \sim G_k} [\max\{a_k', \max_{j \neq k} a_j\}]$.

For each product $i \in [n]$, define

$$x_i(a) = \begin{cases} 1 & a_i > \max\{a_j : j \in [n] \setminus \{i\}\} \text{ and } o_k < w_k^A(a_{-k}), \\ 0 & \text{otherwise.} \end{cases}$$

$$y_i(a) = \begin{cases} 1 & i = k \text{ and } o_k \geq a_k^A(a), \\ 0 & \text{otherwise.} \end{cases}$$

Remark that by the form stability constraint, $a_k$ must not affect the display form of product $k$, so we compare $o_k$ with $w_k^A(a_{-k}) = \mathbb{E}_{a_k' \sim G_k} [\max\{a_k', \max_{j \neq k} a_j\}]$ instead of directly comparing $o_k$ and $a_k$. For the CHANGE-$k$ mechanism, we have

$$\mathrm{OBJ}(\mathcal{M}_k^C) = \underset{a}{\mathbb{E}} \left[ \sum_{i \in [n]} (x_i(a)a_i + y_i(a)o_i) \right]$$

$$= \underset{a}{\mathbb{E}} \left[ \mathbb{I} \left[ o_k \leq w_k^A(a_{-k}) \right] \max_{i \in [n]} a_i \right.$$

$$\left. + \mathbb{I} \left[ o_k > w_k^A(a_{-k}) \right] o_k \right]$$

$$= \underset{a_{-k}}{\mathbb{E}} \left[ \max \left\{ o_k, \underset{a_k}{\mathbb{E}} [\max_i a_i] \right\} \right]. \tag{9}$$

It is easy to verify that each $\mathcal{M}_k^C$ mechanism satisfies the constraints in (5), and the CHANGE mechanism implements the best among these mechanisms.

**Definition 5.8** (CHANGE Mechanism). The CHANGE mechanism, denoted by $\mathcal{M}^C$, runs the mechanism with the highest objective among $\mathcal{M}_1^C, \mathcal{M}_2^C, \cdots, \mathcal{M}_n^C$.

**Theorem 5.9.** *In the case with $n = 2$, $\mathcal{M}^C$ is optimal.*

**Proof of Theorem 5.9.** When $n = 2$, $\mathcal{M}^C$ runs the best mechanism between $\mathcal{M}_1^C, \mathcal{M}_2^C$. We have

$$\mathrm{OBJ}(\mathcal{M}_1^C) = \underset{a_2}{\mathbb{E}} \left[ \max\{o_1, \underset{a_1}{\mathbb{E}} [\max\{a_1, a_2\}]\} \right],$$

$$\mathrm{OBJ}(\mathcal{M}_2^C) = \underset{a_1}{\mathbb{E}} \left[ \max\{o_2, \underset{a_2}{\mathbb{E}} [\max\{a_1, a_2\}]\} \right].$$

Let $x^*(a)$ and $y^*(a)$ be the optimal solution to (5), inducing the optimal mechanism $\mathcal{M}^*$. We have known in the proof of ?? that either $y_1^*(a) \equiv 0$, or $y_2^*(a) \equiv 0$. By symmetry we assume $y_1^*(a) \equiv 0$.

For any $a_1$, $y_2^*(a_1, a_2)$ is independent of $a_2$ by form stability. If $y_2^*(a_1, a_2) = 0$ for all $a_2$, we have $\mathbb{E}_{a_2}[\mathrm{OBJ}^*(a_1, a_2)] \leq \mathbb{E}_{a_2}[\max\{a_1, a_2\}]$. And if $y_2^*(a_1, a_2) = 1$ for all $a_2$, we have $\mathbb{E}_{a_2}[\mathrm{OBJ}^*(a_1, a_2)] = o_2$.

Therefore we have $\text{OBJ}(\mathcal{M}^*) \leq \mathbb{E}_{a_1}[\max\{o_2, \mathbb{E}_{a_2}[\max\{a_1, a_2\}]\}] = \text{OBJ}(\mathcal{M}_2^C) \leq \text{OBJ}(\mathcal{M}^C)$. $\qquad \square$

Similar with Theorem 5.6, we also have a result for Change mechanism when the number of bidders are unrestricted.

**Theorem 5.10.** *For content merging problem with n participants, assume the bid distributions are identical, $\mathcal{M}^C$ is $\frac{n-1}{n}$-competitive relative to the optimal mechanism $\mathcal{M}^*$.*

PROOF OF ??. Let $o_k = o_* = \max_{i \in [n]} o_i$. Clearly, $\mathcal{M}_k^C$ is the optimal mechanism among $\{\mathcal{M}_i^C\}_{i \in [n]}$ and $\mathcal{M}^C = \mathcal{M}_k^C$, and let $o_i \equiv o_*$ will only make the revenue of $\mathcal{M}^{OPT}$ larger instead of $\mathcal{M}^C$. Therefore, without loss of generality, we assume $o_i = o_*$ for all $i \in [n]$.

Let $G$ be the common distribution of $a_i$. Firstly we give an upper bound on the optimal mechanism. Let $x^*(\boldsymbol{a}), y^*(\boldsymbol{a})$ be the optimal solution to problem (5), inducing the optimal mechanism $\mathcal{M}^*$. Let $Rev(\boldsymbol{a})$ be the value of chosen item when the bid profile is $a$.

Given arbitrary bid profile $\boldsymbol{a}$, for arbitrary legal mechanism, one of the following statements must hold.

(1) there exists $i_0$ such that $y_{i_0}(a_{i_0}, a_{-i_0}) = 1$, in this case, for $\forall b_{i_0}, y_{i_0}(b_{i_0}, a_{-i_0}) = 1$
(2) for all $i \in [n]$, $y_i(\boldsymbol{a}) = 0$, in this case, for $\forall b_i, y_i(b_i, a_{-i}) = 0$ (otherwise it will contradict the first statement if we let $\boldsymbol{a} = (b_i, a_{-i})$).

If the first statement holds, then take expectation on $a_{i_0}$, we have

$$\mathbb{E}_{a_{i_0} \sim G}[Rev(a_{i_0}, a_{-i_0})] = o_{i_0} = o_* \qquad (10)$$

If the second statement holds, then for all $i \in [n]$,

$$\mathbb{E}_{a_i \sim G}[Rev(a_i, a_{-i})] \leq \mathbb{E}_{a_i \sim G} \max\{a_i, o_*, a_{-i}\} \qquad (11)$$

Specifically, we can take $i = i_0$.

Let $g(t) = \mathbb{E}_{a \sim G}[\max\{a, t\}]$, since at least one of these two statement holds, we have

$$\mathbb{E}_{a_{i_0} \sim G}[Rev(a_{i_0}, a_{-i_0})] \leq \max\{o_*, a_{i_0}, a_{-i_0}\}$$
$$= \mathbb{E}_{a_{i_0} \sim G} \max\{o_*, g(\max_{k \neq i_0} a_k)\}$$
$$\leq \max_{i \in [n]} \max\{o_*, g(\max_{k \neq i} a_k)\}$$

Take expectation on the full distribution $G^n$, we derive that

$$\mathbb{E}_{\boldsymbol{a} \sim G^n}[Rev(\boldsymbol{a})] \leq \mathbb{E}_{\boldsymbol{a} \sim G^n}[\max_{i \in [n]} \max\{o_*, g(\max_{k \neq i} a_k)\}]$$

and therefore

$$\text{OBJ}(\mathcal{M}^{OPT}) \leq \mathbb{E}_{\boldsymbol{a} \sim G^n}\left[\max_{i \in [n]} \max\{o_*, g\left(\max_{k \neq i} a_k\right)\}\right].$$

When $t > o_*$,

$$\Pr\left[\left(\max_{i \in [n]} \max\left\{o_i, g\left(\max_{k \neq i} a_k\right)\right\} \leq t\right)\right]$$
$$= \Pr\left[\left(\forall i, g\left(\max_{k \neq i} a_k\right) \leq t\right)\right]$$
$$= \Pr\left[\left(\forall i, \max_{k \neq i} a_k \leq g^{-1}(t)\right)\right]$$
$$= \left(F\left(g^{-1}(t)\right)\right)^n$$

Thus, we have

$$\text{OBJ}(\mathcal{M}^{OPT}) \leq o_{k^*} + \int_{o_*}^{\infty}\left(1 - \left(F\left(g^{-1}(t)\right)\right)^n\right) dt$$
$$= o_* + \int_{g^{-1}(o_*)}^{\infty}\left(1 - (F(t))^n\right) g'(t) dt$$
$$= o_* + \int_{g^{-1}(o_*)}^{\infty}\left(1 - (F(t))^n\right) F(t) dt$$

where the last equality holds since $g'(t) = F(t)$.

Next, we consider the change mechanism,

$$\text{OBJ}(\mathcal{M}^C) \geq \text{OBJ}(\mathcal{M}_k^C)$$
$$= \mathbb{E}_{a_{-k}}\left[\max\left\{o_*, \mathbb{E}_{a_k}[\max_i a_i]\right\}\right]$$
$$= o_k + \int_{o_*}^{\infty}\left(1 - \left(F\left(g^{-1}(t)\right)\right)^{n-1}\right) dt$$
$$= o_k + \int_{g^{-1}(o_*)}^{\infty}\left(1 - (F(t))^{n-1}\right) F(t) dt$$

Since $1 - x^n$ is a concave function with $n$, we have $1 - F(t)^{n-1} \geq \frac{n-1}{n}(1 - F(t)^n) + \frac{1}{n}(1 - F(t)^0) = \frac{n-1}{n}(1 - F(t)^n)$. Thus, we finally obtain

$$\frac{OBJ(\mathcal{M}^C)}{OBJ(\mathcal{M}^{OPT})} \geq \frac{n-1}{n},$$

$\qquad \square$

# 6 CONCLUSION

In this paper, we recognize a common situation in designing the merging mechanism of ad and organic items, where ad and organic items can overlap, which potentially causes an incentive problem in almost all existing works. We revisit the mechanism design under this situation, with strict adherence to the economic properties. We provide some characterization results on the mechanism and show a necessary property called form stability. We design two mechanisms, called the FIX mechanism and CHANGE mechanism, and analyze their performance under certain conditions.

For future works, we may explore the following directions: The first is to design mechanisms that are optimal or approximately optimal in the cases with $n > 2$ and asymmetric bid distributions. The second is to extend the current results to more general cases such as multi-slot scenarios. The third is to practically estimate the effectiveness of the proposed mechanisms in real business scenes.

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
