# OpenReview forum: "Ad vs Organic: Revisiting Incentive Compatible Mechanism Design in E-commerce Platforms"
_ACM.org/TheWebConf/2024/Conference — TheWebConf24_

### Official Review · Reviewer_YhnN · 2023-11-19

**Novelty:** 5
**Technical Quality:** 5

**Review:**

Summary of Main Content:
The paper explores an intriguing yet overlooked scenario in existing literature: designing the merging mechanism of ad and organic items, wherein ad and organic items can overlap. The paper formally articulates the mechanism design problem in the aforementioned scenario and simplifies this problem by introducing a necessary condition named form stability. Finally, this paper designs two mechanisms, called the FIX mechanism and CHANGE mechanism, and analyze their competitive ratios under certain conditions

Strengths:
The paper explores the intersection of advertising auctions and recommendation systems, a topic that is novel, intriguing, and has garnered considerable attention. From the introduction of the problem background to the formulation and simplification of the mechanism design problem, and then to the design and analysis of two mechanisms in terms of their performance, the logic is clear, progressively deepened, and easy to comprehend.

Weaknesses:
There are many typos throughout the paper. For example, 'display' is misspelled in the first line, In Definition 5.1, there are two consecutive letters 'j', and there are numerous instances of garbled references to equations, such as in lines 346 and 381. While these issues do not hinder the understanding of the content, they do significantly affect the reader's experience, giving the impression that the authors did not rigorously quality-check the paper before submission

Advices:
The paper is generally well-written and technically sound. The only suggestion I have is to diligently revise and correct the typographical errors. Before submitting in the future, ensure a thorough quality check to prevent such typographical issues from affecting the outcomes and credibility of the paper.

**Questions:**

1.Given the extensive research in integrating ad and organic items in online advertising and recommend systems, how does your work differentiate from these existing approaches, especially in terms of handling the overlap of ad and organic items?
2.Please explain in detail the incentive problems present in existing works, the challenges they pose for mechanism design, and how you have addressed and overcome these issues.
3.Is there any real-world data indicating that the candidate ad and organic items may overlap, thereby validating the practical relevance of the research topic addressed in this paper?

**Reviewer Confidence:**

3: The reviewer is confident but not certain that the evaluation is correct

**Scope:**

4: The work is relevant to the Web and to the track, and is of broad interest to the community

---

### Official Review · Reviewer_9yLW · 2023-11-21

**Novelty:** 6
**Technical Quality:** 6

**Review:**

**Summary:**
The paper discusses ad auctions where organic and paid impressions may overlap. This presents a challenge to the platform that wishes to maximize both revenue and the quality of its recommendation system (user experience), and requires a rethinking of incentive-compatibility for the advertisers. The paper studies the case of a single slot. It characterizes possible mechanisms following Myerson's analysis of the single-item auction, and then presents two content-merging mechanisms and proves their (approximate-)optimality in different cases.

**Strengths**
- The motivating idea of the paper is very nice, relevant, and grounded in realistic settings.
- The analysis seems well-founded, and the suggested mechanisms yield good results.

**Weaknesses**
- No proof of Lemma 5.5 in the appendix.

**Minor comments**
- Several undefined references, e.g., line 346, line 381, line 807.
- Please include a reference to the 236.90 billion USD statistic.
- Line 158: "FIX is 4/5-competitive relative to the optimal objective" - what optimal objective?
- Line 160: "and all bid distributions are identical" - and independent?
- Line 221: "which is a standard assumption in economics" - refer to Myerson '81
- Line 253: You refer to X_i as the total -probability-, but this is before you make the feasibility requirement (line 285) that guarantees it is indeed a probability.
- Line 256: "The incentive compatibility requires" - rephrase.

**Questions:**

- Line 415: "and this is basically equivalent to the following" - Do you mean that Lemma 4.4 and Lemma 4.5 are equivalent? Unclear.
- An important part of Myerson’s result is that the highest revenue DSIC mechanism is also the highest revenue BIC mechanism. Is that also the case here? It seems possible that the platform could convince the product owner to pay for an ad in a Bayesian fashion, even when it should be organically displayed, and this may help the objective.

**Reviewer Confidence:**

3: The reviewer is confident but not certain that the evaluation is correct

**Scope:**

4: The work is relevant to the Web and to the track, and is of broad interest to the community

---

### Official Review · Reviewer_VU33 · 2023-11-24

**Novelty:** 4
**Technical Quality:** 3

**Review:**

The paper studies an ad-auction problem, where a product can be displayed to the user either as an organic item or as an ad item. In particular, the authors focus on the single-slot scenario under this setting, and their goal is the design of mechanisms that are truthful, individually rational, and provide good approximation guarantees in terms of revenue and user experience.  Among others they characterize the content merging mechanisms in this setting, and also design two truthful mechanisms and analyze theoretically their performance.

The introductory sections of the paper, in my opinion, are not that well-written. I have read the intro part many times and I was not able to understand what is the problem that the paper studies or how the problem is motivated (as the description of the real-life application is not clear).  The same goes for the preliminaries section and the presentation of the model that is not that formal. All these sections should be restructured and rewritten. Technically the results that the paper presents are not trivial, but on the other hand the approaches that are followed are kind of standard (e.g., adaptations of Myerson’s lemma etc.).

Overall the paper tries to capture and analyze theoretically some real life problems in ad-auctions, but mostly due to the way that it is written I am not sure if it can be considered for publication.

**Questions:**

None.

**Reviewer Confidence:**

2: The reviewer is willing to defend the evaluation, but it is likely that the reviewer did not understand parts of the paper

**Scope:**

3: The work is somewhat relevant to the Web and to the track, and is of narrow interest to a sub-community

---

### Official Review · Reviewer_zCf5 · 2023-11-25

**Novelty:** 5
**Technical Quality:** 5

**Review:**

Summary:

This paper studies the auction design problem when a product can be displayed in two possible forms of an ad item or an organic item. Building on top of the classic Myerson's lemma, the authors provide characterizations of truthful auctions and demonstrate an interesting property of form stability. The authors further propose two mechanisms, FIX and CHANGE, that are approximately optimal when there are two bidders or bidders are identical.

Comments:

This paper is generally well-written, clear, and easy to follow. The problem studied in the paper is interesting and relevant to the auction design in online advertising. The characterizations are clean and elegant while the proposed mechanisms are simple but effective.

1. The characterizations are neat but not deep or surprising as they are mostly based on the classic Myerson's lemma.

2. It seems that the authors implicitly restrict their attention to deterministic mechanisms. It would be interesting to see whether results/characterizations can be generalized to randomized mechanisms.

3. More justification are needed for the assumption that the organic result provides better click-through rate than an ad item.

4. It seems that the proof of Lemma 4.4 depends on assumptions mentioned in 2 and 3 above. Do they continue to hold without these assumptions?

5. Proofs are needed to show that FIX and CHANGE are truthful as mechanisms are combined by taking a max (while usually, mechanisms are combined via randomization, which maintains truthfulness). In particular, it seems that FIX might not be truthful: it looks like it is possible that the winners are the same for all k but the payments are different.

6. The approximation results only apply to restricted settings in which there are two bidders or bidders are identical. The paper would be stronger if the approximation results can be extended to more general settings.

**Questions:**

Could the authors comment in questions mentioned in the review above?

**Reviewer Confidence:**

4: The reviewer is certain that the evaluation is correct and very familiar with the relevant literature

**Scope:**

3: The work is somewhat relevant to the Web and to the track, and is of narrow interest to a sub-community

---

### Official Review · Reviewer_dJeo · 2023-11-29

**Novelty:** 6
**Technical Quality:** 6

**Review:**

The paper considers a very interesting problem on the advertisers' incentive of organic vs ad presence of their items --- I have constantly encountered such kind of problem when I am doing
search on google & amazon, that an item is promoted while also being top results organically. The paper formulates this problem as a mechanism design problem, and seeks for the truthful mechanism that maximizes a combination of ad revenue and user experience. The paper explicitly construct several natural mechanisms that approximate the optimal objective under symmetric assumptions. Overall, I would like to recommend acceptance of this paper for the cute problem studied in this paper and its clean solution styles. That said, the paper still have some issues that need to be addressed before acceptance. For example, there are several typos in this paper ("dislayed" in the abstract, "Proof of ???" in Theorem 5.10). These issues make me worry about the other potential issues in its technical proofs, but I unfortunately do not have time to check them so I recommend the authors to carefully self-check these parts in the draft as well.

**Questions:**

To what extent, do you think the proposed mechanism can be applied to the multi-slot case where a ranked list of items (possible ad or organic) is displayed to the user?

**Reviewer Confidence:**

3: The reviewer is confident but not certain that the evaluation is correct

**Scope:**

4: The work is relevant to the Web and to the track, and is of broad interest to the community

---

### Decision · Program_Chairs · 2024-01-22

**Decision:**

Accept

**Comment:**

The paper studies the practically well-motivated problem of the incentive issues that arise in merging candidates selected for organic results and sponsored results in e-commerce platforms. There is clear consensus about the importance of carefully studying this. The paper formulates a clean theoretical model of this setting and provides characterizations of truthful mechanisms, which are clean, but not too deep. The approximation results for the simple and nice mechanisms constructed in this work apply to quite a restrictive setting of two bidders, or when bidders are identical, and the results are for the single-slot case. Overall, the results in this paper are worth publishing to hopefully stimulate further study on this topic, even if the paper is not too strong.